# Using Healthcare Complaints Analysis Tool to Evaluate Patient Complaints during the COVID-19 Pandemic at a Medical Center in Taiwan

**DOI:** 10.3390/ijerph20010310

**Published:** 2022-12-25

**Authors:** Shu-Chuan Wang, Nain-Feng Chu, Pei-Ling Tang, Tzu-Cheng Pan, Li-Fei Pan

**Affiliations:** 1Department of Medical Affair Administration, Kaohsiung Veterans General Hospital, Kaohsiung City 813414, Taiwan; 2Division of Occupational Medicine, Kaohsiung Veterans General Hospital, Kaohsiung City 813414, Taiwan; 3School of Public Health, National Defense Medical Center, Taipei City 11490, Taiwan; 4Research Center of Medical Informatics, Kaohsiung Veterans General Hospital, Kaohsiung City 813414, Taiwan; 5General Affairs Administration, Kaohsiung Veterans General Hospital, Kaohsiung City 813414, Taiwan

**Keywords:** patient complaints, hospital environment, patient communication, COVID-19 pandemic, HCAT

## Abstract

The purpose of this study is to evaluate patient complaints using the Healthcare Complaints Analysis Tool (HCAT) during the COVID-19 pandemic in 2021 in Taiwan. Additionally, the study examines the distribution and type of patient complaints before and during the COVID-19 pandemic to provide a better clinical procedure, hospital management and patient relationship. This study utilizes a cross-sectional design. We collected patient complaints from January 2021 to December 2021 at a medical center in Southern Taiwan. Using the Healthcare Complaints Analysis Tool (HCAT), the patient complaints are classified and coded into three major domains (clinical, management and relationship), and seven problem categories (quality, safety, environment, institutional process, respect and patient rights, listening and communication). We further compared and categorized the complaints based on whether they were COVID-19-related or not and whether it was before or during the COVID-19 pandemic to understand the differences in patient complaints. In total, we collected 584 events of patient complaints. Based on the HCAT domains, the complaints about management were the highest, at 52.9%, followed by complaints about relationship, about 37.7%. According to the types of problem, the complaints about the environment were the highest, about 32.5% (190/584), followed by communication at about 29.6% (173/584), and institutional process at about 20.4% (119/584). There were 178 COVID-19-related complaints and they were made more frequently during Q3 and Q4 (from mid-June to December) which was the pandemic period in 2021 in Taiwan. Among the COVID-19-related complaints, the most frequent were in the environment domain with 114 cases (about 65.7% of COVID-19-related complaints). The domains of patient complaints were statistically different between COVID-19-related and non-related (*p* < 0.001). During the COVID-19 pandemic, the proportion of COVID-19-related complaints increased 1.67 times (117/312 vs. 61/272, *p* < 0.001). Both prior to and during the COVID-19 pandemic, management-related complaints represented the highest domain. During the COVID-19 pandemic, the implementation of infectious disease prevention and control policies and actions may have developed some inconvenience and difficulty in seeking medical practice and process. These characteristics (complaints) are more prominent, and timely and patient-first consideration is required immediately to build up better clinical procedures, the healthcare environment and comprehensive communication. Using the HCAT can allow health centers or health practitioners to understand the needs and demands of patients through complaints, provide friendly medical and health services, avoid unequal information transmission, build trust in doctor–patient relationships and improve patients’ safety.

## 1. Introduction

The outbreak of COVID-19 has influenced healthcare practice procedures and regulations in many ways since 2020. During the COVID-19 pandemic, some new healthcare policies and regulations were implemented to control the outbreak of such a serious infectious disease [1,2]. However, these policies and rules may have made patients feel quite troubled in the process of seeking medical procedures and the patient complaints increased significantly during the pandemic in 2021. 

Patient complaints can provide valuable insights regarding the quality and safety of clinical practice, as they can act as independent assessors of the healthcare service and often reflect the expectations of society as a whole [3]. Healthcare complaints are an underused source of data for augmenting existing monitoring tools and, more importantly, patient complaint surveys have been shown to reflect the quality of health and to ensure patients’ safety, so can be used as a tool to monitor patients’ safety and satisfaction with healthcare administration [4]. Healthcare complaints conventionally refer to an expression of grievance and dispute, typically written and communicated through a letter by a patient or their family, about the receipt of healthcare [5]. One way to improve the quality of healthcare is to systematically analyze the complaints sent in by patients or their relatives and then identify the areas or situations of the complaints. This approach expands the perspective of quality improvement to encompass patient reports of care experiences [6].

The healthcare complaint analysis tool (HCAT) was defined by Gillespie et al., (2015) by aggregating the coding taxonomies from studies included in the systematic review, revealing 729 uniquely worded codes that were then refined and conceptualized into three broad domains and seven categories [4]. It provides a reliable tool for coding complaints and measuring the severity of complaints, to facilitate service monitoring and organizational learning [4,5].

This study uses the HCAT to evaluate patient complaints before and during the COVID-19 pandemic in 2021 in Taiwan. Additionally, examining the distribution and type of patient complaints before and during the COVID-19 pandemic can provide better hospital management and patient relationships in the future.

## 2. Materials and Methods

### 2.1. Study Population

This study used a cross-sectional design; patient complaints were collected from January 2021 to December 2021 from a medical center in Southern Taiwan. Patient complaints were collected from different channels including the patient relations office, external letters, oral opinions, public reviews, suggestion boxes, superintendent’s mail box, letters from the competent authority and E-mails. All complaints were collected by the patient relations office of the hospital and checked by the staff daily. In total, 584 complaints were included in this study.

### 2.2. Study Time Frame

The health complaints were collected from January 2021 to December 2021, due to the outbreak of COVID-19 that occurred in the middle of June in Taiwan. So, we divided the study time frame as Q1–Q2 and Q3–Q4 which are from January to mid-June and mid-June to December, respectively, as before and during the COVID-19 pandemic in Taiwan.

### 2.3. Approval of Study Committee

The study data are all anonymous and come from different channels. In order to analyze and interpret these data, we proposed using these data to the hospital quality management committee and the analysis was approved by this committee. 

### 2.4. Healthcare Complaints Coding

The patient complaint events were coded using the HCAT method, and are classified into three major domains (clinical, management and relationship), and seven problem categories (quality, safety, environment, institutional process, respect and patient rights, listening and communication). The coding was completed independently by two experienced technicians. Any differences in coding were further discussed, then decided upon. Regardless of the original channels of the complaints, coding of all the complaints and problems in this study was exclusively based on the text records from the file database. While the majority of complaints pertained to overall care, some complaints focused on specific aspects of care.

### 2.5. Statistical Methods

We used SPSS ver. 22 (IBM Corp., Armonk, NY, USA) to conduct all statistical analyses. The categorical variables were described as numbers and percentages. Chi-square tests were used to compare the differences among groups. A two-tailed p-value of less than 0.05 was considered statistically significant. We further compared the distribution and type of patient complaints as to whether they were COVID-19-related or non-related at the time of the COVID-19 pandemic (before vs. during) to understand the impact of the COVID-19 pandemic on patient complaints. 

## 3. Results

Table 1 presents the distribution of the HCAT domains and problems. Regarding the domain, among these 584 complaints, management-related complaints were the most prevalent (309 events; (52.9%)), followed by relationship complaints (220 events; (37.7%)). According to the types of problem, the most frequent were environment complaints with 190 events (32.5%), followed by communication complaints, which had 173 events (29.6%), and institutional process complaints, which had 119 events (20.4%). In terms of environment, the physical environment had the most prevalent complaints with 123 events (21.1%). For institutional process, the length of waiting lists had the highest complains with 78 events (13.4%). The highest complaint problem for communication was poor spoken language, which was 123 events (21.1%).

Table 2 shows the distribution of the HCAT domains and problems based on whether they are COVID-19-related or not. In Q3 and Q4 of 2021, there were 117 COVID-19-related complaints. Among the total COVID-19-related complaints, the most frequent were in the management domain with 114 cases (114/178, about 64.0%). The problem domains of patient complaints were significantly different between COVID-19-related and non-related events (*p* < 0.001). 

The HCAT domains and problems before and during the COVID-19 pandemic period are shown in Table 3. There was no statistical difference in medical staff (nurse vs. non-nurse) before and during the COVID-19 pandemic. However, during the COVID-19 pandemic, the proportion of COVID-19-related complaints increased by 1.67 times (117/312 vs. 61/272, *p* < 0.001). Both prior to and during the COVID-19 pandemic, management-related complaints represented the highest domain.

Figure 1 demonstrates the distribution of COVID-19-related HCAT domain complaints (n = 178) before and during the COVID-19 pandemic. Before and during the COVID-19 pandemic, management complaints were the most frequent domain, which had about 65.5% (40/61) and 63.3% (74/117) of complaints, respectively. 

## 4. Discussion

In this study, we found that patient complaints may have changed during the COVID-19 pandemic. Overall, the most frequent domain of patient complains was management with 309 events, followed by relationship with 220 events. Regarding the complaint category, environment had the highest complaints with 190 events, followed by communication, 173, and institutional process, 119 events. During the COVID-19 pandemic, patient complaints slightly increased from 46.6% to 53.4%. Furthermore, the complaints of COVID-19-related events increased significantly, especially in the management domain.

Healthcare complaints conventionally refer to an expression of grievance and dispute, typically written and communicated through a letter by a patient or their family, about the receipt of healthcare [7]. Jerng et al. indicated that the patients and families used multiple channels to issue complaints, such as through feedback sheets, in addition to the traditional complaint process [5]. It might be difficult to distinguish between comments and negative feedback, complaints and other informal comments such as telephone calls, threats of lawsuits and comments to caregivers commonly provided to the hospital as the sources for the collection of patient feedback [8].

Complaints occur when a threshold of dissatisfaction has been breached, with dominant motivations being to correct an ongoing problem or prevent recurrence [9]. Other themes for improvement reported in the literature included ICU atmosphere, amenities for visiting relatives, emotional support, consistency, clarity and completeness of information, respect and compassion towards families, the inclusion of family and support during decision-making processes [5,10]. 

In terms of complaint content, a systematic review which included 59 primary studies reported that approximately one third of complaints relate to the safety and quality of clinical care, one third to the management of the healthcare organization and one third to healthcare staff and patient communication [3,11]. Healthcare complaints have been shown to reveal problems in patient care (such as medical errors, breaching clinical standards and poor communication) not captured through safety and quality monitoring systems (such as incident reporting, case review and risk management) [4,12,13,14]. When a patient makes a formal complaint, a threshold of dissatisfaction has been reached, and such complaints have been shown to highlight deficiencies in the quality and safety of healthcare [3]. In our data, most complaints are related to the management domain.

The patient complaints might actually precede, rather than follow, safety incidents, potentially acting as an early warning system [4,11]. As awareness and knowledge about healthcare has increased, demand from the public and individuals has surged. Likewise, complaints from patients could serve as a management tool for continuous quality improvement [15]. Since patients’ complaints are difficult to avoid in most clinical practices, reports recommend that healthcare institutions include the analysis of complaints as a possible measurement of service quality [15,16]. This also requires systematic procedures for analyzing the complaints, as is the case with adverse event data [17]. While complaints have typically been considered in terms of risk management, they can also be an opportunity to gain an insight into patient perceptions of safety in healthcare. Patients are in a position to identify areas of risk within healthcare that are not discernible by staff. Therefore, incorporating patient insights into quality improvement can complement other measurement and monitoring tools [18,19]. In past research, healthcare complaints focused on complaint handling, physician behavior, high-risk clinicians and malpractice claims [20,21,22,23]. However, studies increasingly suggest that patient experience reflects care quality [24]. Whether complaints contain useful information is no longer the focus, but rather how valid insights can be reliably extracted [9].

Many complaints aim to contribute information that will improve healthcare delivery [4,25]. The HCAT is capable of reliably identifying the problems, severity, stage of care and harm reported in healthcare complaints [4]. It also has the ability to reliably code severity within each complaint category. The HCAT provides a reliable additional data stream for monitoring healthcare safety and quality [4,26].

The HCAT offers one way of characterizing patient complaints, and its use is gaining ground [3,4,6,9,11]. The developers of the HCAT used the tool to investigate areas of hot spots and blind spots. Hot spots refer to areas where harm and near-misses accumulate, and blind spots refer to aspects of care that are difficult to monitor in healthcare complaints in the UK [6]. When systematically analyzed, and considered through a quality improvement rather than a risk management lens, seven healthcare complaints had the potential for identifying hot spots (as points in care with a high prevalence of harm or near-misses) and blind spots (as points in care that cannot be observed by staff members) [9,18].

In the UK, relationship problems were the most prevalent event at around 40%~53% in total patient complaints [18,27,28]. Among the relationship domain, 58.6% were related to respect and patient rights, 25.9% were related to listening and 15.5% were related to communication [18]. Data from the Irish Health Service Executive and Irish Medical Indemnity Company showed that a total of 230 complaints, encompassing 432 issues and complaints, were categorized. Relationship issues emerged the most frequently (40%) [18]. The HCAT coding in another single-center, retrospective study conducted of patients surgically treated for a chronic subdural hematoma between October 2014 and January 2019 showed that 53% were relationship problems, 40% were management problems and 7% were other problems [28]. For the relationship problems, 75% were classified as problems with communication and 25% as problems with listening. Among the communication problems, they could be further divided into lack of communication (50%), and incorrect (33%) and delayed (17%) communication [28]. These results were slightly different from our findings in that relationship problems were the second most prevalent. In a case–control study, the complaints were collected from the relatives of patients who died in the hospital between January 2015 and December 2017 in three district general hospitals in the UK. About fifty five percent of complaint items were related to quality and safety issues, forty percent were related to relationship issues, such as lack of humaneness and care, and only four percent were related to management issues [27].

In Denmark, clinical problems were the most frequent event which were around 83.9~97.5% [6,29]. Furthermore, in data from 712 patient compensation cases containing 1305 complaint points that related to emergency care, the most prevalent problems were also related to clinical domain (83.9%) [6]. A sample of 1,613 compensation claims to the Danish Patient Assurance Organization was reviewed using a standardized taxonomy in the clinical domain covering healthcare quality and patient safety issues. They observed a baseline increase in claims relating to clinical problems, but this increase was less pronounced following a reorganization. However, diagnostic errors and patient outcomes showed an insignificant tendency to increase [29]. Furthermore, in data from 712 patient compensation cases containing 1,305 complaint points related to emergency care, the most problems were also related to clinical domain (83.9%).

In Taiwan, a retrospective study of the complaints of a university-affiliated medical center was conducted from 2008 to 2016. A total of 1529 problems (441 from the ICU and 818 from the general ward) were collected. Management problems were the most frequent complaint amongst the ICU and general ward, which were about 44.7%. ICU management problems had more environment-related complaints than the general ward. Furthermore, compared to the general ward, ICU relationship problems were about communication and less listening [5]. This is similar to our study in that the management problems were the most common. We further compared the patterns of patient complaints in different countries. In the UK, relationship problems were the most frequent complaint, but the most frequent complaints in Denmark were about clinical problems. However, these event sources were limited to single patients, such as surgical or emergency, from service executive or insurance organizations. In addition, results from Taiwan have shown that the management problems were the most frequent complaints, which is different to other countries.

Furthermore, our data were from the first study using the HCAT to evaluate patient complaints before and during the COVID-19 pandemic in Taiwan. Whether during the COVID-19 pandemic or not, the management domain still had the most complaints, at about 52.9%. This may be due to different doctor–patient relationship culture in Taiwan. We have a very comprehensive national health insurance system and patients can visit medical centers easily, and there is also very high loyalty to medical centers. So, even as the COVID-19 pandemic developed, management complaints were still the most prevalent. The implementation of infectious disease prevention and control policies and actions during the COVID-19 pandemic period may have developed some inconvenience with seeking medical care.

Our study has some limitations. First, the cross-sectional study without follow-up limits our ability to evaluate the causal relationships between COVID-19 and patient complaints. Second, we did not collect other demographic characteristics and disease severity in the study. The relationship between complainer and patient was unclear. Third, our questionnaire did not distinguish between the severity of complaints as low, medium and high. However, this would not affect the distribution of patient complaints in our study. Finally, there is potential misclassification bias for the coding of patient complaints. However, this misclassification might be non-differential which may dilute the significance of our results.

## 5. Conclusions

The HCAT can contribute to the management of healthcare and can integrate existing complaints into the HCAT severity ratings that can then be extracted and passed onto managers, external monitors and researchers. The HCAT could be used as an alternative metric of success in meeting standards (for example, on hospital hygiene, waiting times and patient satisfaction). It could also be used to follow up on clinical, management or relationship interventions.

During the COVID-19 pandemic, the implementation of infectious disease prevention and control policies and actions may have developed some inconvenience and difficulty in seeking medical practice and processes. A failure to meet the demands and expectations of patients may increase psychological stress on the patients and distort the doctor–patient relationship. During the pandemic, these characteristics (complaints) were more prominent. Solving these problems requires immediate, timely and patient-first consideration to build up better clinical procedures, healthcare environment and comprehensive communication, such as the moving line process, the accompanying policy for in-patient services and vaccine administration. Using the HCAT helps the hospital to understand the needs and demands of patients through complaints, provide friendly medical and health services, avoid unequal information transmission, build trust and quality in the doctor–patient relationship and improve patients’ safety and satisfaction.

## Figures and Tables

**Figure 1 ijerph-20-00310-f001:**
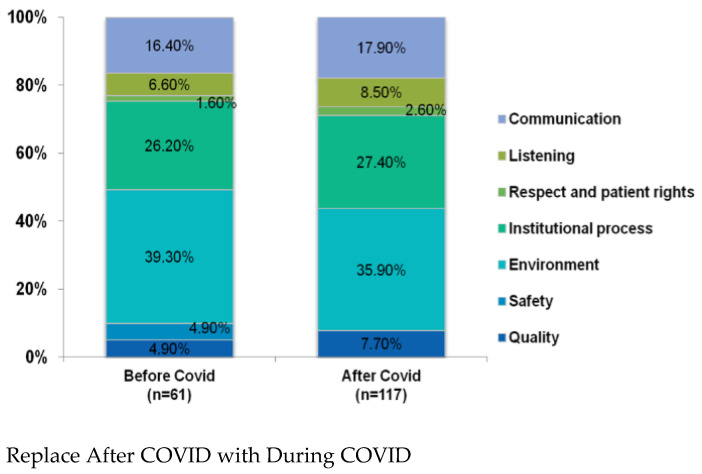
The healthcare complaints analysis tool domains and problems with COVID-19-related events before and during the COVID-19 pandemic (n = 178).

**Table 1 ijerph-20-00310-t001:** Distribution of patient complaints using the healthcare complaints analysis tool domains and problems (n = 584).

Problem Domain	Problem Category	Problem Type	n
Clinical			
(n = 55)	Quality		
	(n = 51)	Inadequate patient assessment	4
		Failure to supervise or monitor care	4
		Unsatisfactory treatment	38
		Problems with records	5
	Safety		
	(n = 4)	Wrong patient or body part	2
		Misdiagnosis	2
Management			
(n = 309)	Environment		
	(n = 190)	Physical environment	123
		Telephone system	6
		Poor administration	57
		Inadequate disposal of drugs	4
	Institutional process		
	(n = 119)	Length of NHS waiting lists for treatment	78
		Surgery cancelling appointments	4
		Patient access to care	10
		Cost	27
Relationship			
(n = 220)	Respect and patient rights		
	(n = 11)	Alleged assault	4
		Impolite behavior	2
		Breach of confidentiality	3
		Discrimination	2
	Listening		
	(n = 36)	Not taken seriously	8
		Unmet patient expectations/requests	28
	Communication		
	(n = 173)	Inadequate explanation	42
		Poor explanation of illness and of prescription	4
		Inadequate explanation of diagnosis or management plan	4
		Poor spoken English	123

**Table 2 ijerph-20-00310-t002:** Analysis of the healthcare complaints analysis tool domains and problems according to COVID-19-related and non-related events.

	Total	COVID-Related	Non-COVID-Related	
Variables	(N = 584)	(n = 178, 30.5%)	(n = 406, 69.5%)	*p **
	n (%)	n (%)	n (%)	
Medical staff				0.07
Nurse	82 (14.0)	18 (10.1)	64 (15.8)	
Non-nurse	502 (86.0)	160 (89.9)	342 (84.2)	
Season period				<0.001
Q1	126 (21.6)	20 (11.2)	106 (26.1)	
Q2	146 (25.0)	41 (23.0)	105 (25.9)	
Q3	154 (26.4)	59 (33.1)	95 (23.4)	
Q4	158 (27.1)	58 (32.6)	100 (24.6)	
Season period				<0.001
Q1 + Q2	272 (46.6)	61 (34.3)	211 (52.0)	
Q3 + Q4	312 (53.4)	117 (65.7)	195 (48.0)	
Problem domain				0.001
Medical	55 (9.4)	15 (8.4)	40 (9.9)	
Management	309 (52.9)	114 (64.0)	195 (48.0)	
Relationship	220 (37.7)	49 (27.5)	171 (2.1)	

* Chi-square test was used to compare the difference between COVID-related and non-COVID-related.

**Table 3 ijerph-20-00310-t003:** Analysis of the healthcare complaints analysis tool domains and problems before and during the COVID-19 occurrence period.

	Total	Before COVID-19	During COVID-19	
Variable	(N = 584)	(n = 272, 46.6%)	(n = 312, 53.4%)	*p **
	n (%)	n (%)	n (%)	
Medical staff				0.50
Nurse	82 (14.0)	41 (15.1)	41 (13.1)	
Non-nurse	502 (86.0)	231 (84.9)	271 (86.9)	
COVID-19-related event				<0.001
Yes	178 (37.5)	61 (22.4)	117 (37.5)	
No	406 (69.5)	211 (77.6)	195 (62.5)	
Problem domain				0.41
Medical	55 (9.4)	30 (11.0)	25 (8.0)	
Management	309 (52.9)	144 (53.0)	165 (52.9)	
Relationship	220 (37.7)	98 (36.0)	122 (9.1)	
Relationship	220 (37.7)	49 (27.5)	171 (2.1)	

* Chi-square test was used to compare the difference between before and during the COVID-19 occurrence period.

## Data Availability

Please request data from S.-C. Wang.

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
