# Peer review of "Using Healthcare Complaints Analysis Tool to Evaluate Patient Complaints during the COVID-19 Pandemic at a Medical Center in Taiwan"

_ijerph, 2022, doi:10.3390/ijerph20010310_

Round 1

Reviewer 1 Report

Dear author. Thanks for your contributions.

The attention to the complaints of the patients treated in health centers is an indicator of quality and a fundamental feedback channel for the managers of the care units.

However, I find different aspects to improve:

- I do not appreciate the explicit agreement on the ethical treatment of the information contained in the consulted complaints.

- The presentation of the data in table 1 is somewhat confusing; please adjust it.

- The discussion of the results of your research could have been implemented with more updated documentation, since it is only done briefly.

- The conclusions that it publishes do not expressly reflect the results and objectives of the investigation; should modify the wording of the conclusions, avoiding contributions and assessments not obtained in their results.

Author Response

Dec. 20, 2022

Response to Reviewer 1 Comments

Dear author. Thanks for your contributions.

The attention to the complaints of the patients treated in health centers is an indicator of quality and a fundamental feedback channel for the managers of the care units.

However, I find different aspects to improve:

- I do not appreciate the explicit agreement on the ethical treatment of the information contained in the consulted complaints.

Thank you for your valuable comments. The study data all are anonymous and come from different channels, such as patient relations office, external letters, oral opinions, public reviews, suggestion box, superintendent’s mail box, letters from the competent authority and E-mails. In order to analyze and interpret these data, we proposed to the hospital quality management committee to use these data and the analysis were approved by this committee. We added and revised this in our updated manuscript.

- The presentation of the data in table 1 is somewhat confusing; please adjust it.

Thank you for your precious comment. There is a little confusion over the percentage and number in the Table 1. We deleted the percentage and retained the number only. We revised and reformatted this Table in our updated manuscript. Thanks again.

- The discussion of the results of your research could have been implemented with more updated documentation, since it is only done briefly.

Thank you for your valuable comments. We added more discussion about types and contents of complaints before and during the COVID-19 pandemic. We also compared the difference between other study or hospital information. We added and revised this in our updated manuscript. Thanks.

- The conclusions that it publishes do not expressly reflect the results and objectives of the investigation; should modify the wording of the conclusions, avoiding contributions and assessments not obtained in their results.

Thank you for your valuable comments. We deleted the contributions and assessments not obtained from our results and only presented the results from our study. We added and revised this in our updated manuscript.

Reviewer 2 Report

Dear authors, 

First, I hope you are doing well. Please find below my comments to you manuscript.

English use must be revised.

Methods

There is no mention of ethics in the methods. How was patient data protected? Did the study need an ethics review board?

Methods need to be further explained to avoid so much confusion in the results section and allow replicability of your results.

Results

How are percentages calculated in table 1?

Please, define previously in methods what Q3 and Q4 stand for (even though it could be inferred, you have to explain it previously). Also, if you are going to analyze according to time frames, you should mention it in the methods.

COVID-related complaints (178) since March 2020? Only those two periods?

I have a confusion. In the abstract, you mention “complaints from Jan 2011 to Dec 2021”. That’s more than 10 years of data. In the methods you mention, 2021 to 2022, what is the real-time frame?

Following the previous question, if indeed you have data since 2011. How can this be compared with only 1 year of COVID-19 complaints? That chi-square could be misleading. You are comparing 2011-2020 complaints with only 1 year of COVID-19 complaints. If you only make the analysis based on frequencies, this could be misleading.

How many of the “After COVID” complaints are COVID related? Again, this could be misleading to the previous table (Table 2). Again, how could there be “COVID-related complaints” before COVID? If COVID did not exist? This is confusing. You even include an “increased 1.67 times” but how? If, before 2020 there was no COVID pandemic?

Discussion

During the COVID-19 pandemic, the patient complaints shift from 46.6% to 35.4.” Sorry, but is hard to follow. Patient complaints “decrease” even though you have stated in your results the opposite.

The limitations paragraph should be at the end, right before “conclusions”.

Minor comments:

Please revise the correct use of abbreviations. For example, in the abstract, you first mention “(HACT)” and then “HCAT”. 

Author Response

Dec. 20, 2022

Response to Reviewer 2 Comments

Dear authors, 

First, I hope you are doing well. Please find below my comments to your manuscript.

Thank you, we are surviving from the COVID-19.

English use must be revised.

Thank you for your comments. We revised our English to make it more readable. Thanks again.

Methods

There is no mention of ethics in the methods. How was patient data protected? Did the study need an ethics review board?

Thank you for your valuable comments. The study data all are anonymous and come from different channels, such as patient relations office, external letters, oral opinions, public reviews, suggestion box, superintendent’s mailbox, letters from the competent authority and E-mails. In order to analyze and interpret these data, we proposed to the hospital quality management committee to use these data and the analysis were approved by this committee. We added and revised this in our updated manuscript.

Methods need to be further explained to avoid so much confusion in the results section and allow replicability of your results.

Thank you for your comments. We added and revised the section of material and method to make this point clearer in the updated manuscript.

Results

How are percentages calculated in table 1?

Thank you for your precious comment. There is a little confusion over the percentage and number in Table 1. We deleted the percentage and retained the number only. We revised and reformatted this Table in our updated manuscript. Thanks again.

Please, define previously in methods what Q3 and Q4 stand for (even though it could be inferred, you have to explain it previously). Also, if you are going to analyze according to time frames, you should mention it in the methods.

Thank you for your precious comment. The outbreak of COVID-19 in Taiwan started in June 2022. So, Q3 and Q4 represents mid-June to Sep. and Oct. to Dec. 2022, respectively. We added and revised the method section to make this point clearer. Thanks again.

COVID-related complaints (178) since March 2020? Only those two periods?

I have a confusion. In the abstract, you mention “complaints from Jan 2011 to Dec 2021”. That’s more than 10 years of data. In the methods you mention, 2021 to 2022, what is the real-time frame?

Following the previous question, if indeed you have data since 2011. How can this be compared with only 1 year of COVID-19 complaints? That chi-square could be misleading. You are comparing 2011-2020 complaints with only 1 year of COVID-19 complaints. If you only make the analysis based on frequencies, this could be misleading.

Thank you for your precious comment. There is a typo of the original manuscript. The health complaints were collected from Jan. 2021 to Dec. 2021. We are sorry that we did not found this mistake that makes the reviewer confused. We revised the time frame, from Jan. 2021 to Dec. 2021, to make the section of method clearer. We added and revised this in our updated manuscript. Thanks again.

How many of the “After COVID” complaints are COVID related? Again, this could be misleading to the previous table (Table 2). Again, how could there be “COVID-related complaints” before COVID? If COVID did not exist? This is confusing. You even include an “increased 1.67 times” but how? If, before 2020 there was no COVID pandemic?

Thank you for your valuable comments. In Taiwan, the COVID-19 became more severe after June 2021. Before this time, there are only few cases of COVID-19 reported. So, from the mid of June to Dec. 2021 (as the Q3 and Q4 of 2021), there are 117 COVID-related complaints during the COVID-19 pandemic (as 117/312). Before COVID-19 pandemic (as the Q1 and Q2 of 2021), there are only 61 COVID-related complaints (61/272). That’s why we said the proportion of COVID-related complaints increased 1.67 times during COVID-19 pandemic (as 117/312 divided by 61/272). We added and revised this to make it clearer. Thanks again.

Discussion

“During the COVID-19 pandemic, the patient complaints shift from 46.6% to 35.4.” Sorry, but is hard to follow. Patient complaints “decrease” even though you have stated in your results the opposite.

Thank you for your comments. We are sorry that there is a typo in our original manuscript. In Table 3, before COVID-19, there were 272 patient complaint events (272/584 = 46.6%) and there were 312 events (312/584=53.4%) during COVID-19 pandemic. So the complaints slightly increased during COVID-19 pandemic. We added and revised this to make it clearer. Thanks again.

The limitations paragraph should be at the end, right before “conclusions”.

Thank you for your comments. We revised this in the updated manuscript. Thanks.

Minor comments:

Please revise the correct use of abbreviations. For example, in the abstract, you first mention “(HACT)” and then “HCAT”. 

Thank you for your precious comment. We corrected the HACT to HCAT in the updated manuscript. Thanks.

Round 2

Reviewer 1 Report

Thanks for your effort.

A substantial improvement is appreciated in the writing of the work, the results and the discussion.

My initial comments have been suitably satisfied.

Thank you

Reviewer 2 Report

Dear authors, 

Thank you for addressing reviewers' suggestions. Happy holidays.